# Metagenomic analysis of the gut microbiota in piglets either challenged or not with enterotoxigenic *Escherichia coli* reveals beneficial effects of probiotics on microbiome composition, resistome, digestive function and oxidative stress responses

**Prasert Apiwatsiri**[1], **Pawiya Pupa**[1], **Wandee Sirichokchatchawan**[2], **Vorthon Sawaswong**[3,4], **Pattaraporn Nimsamer**[4], **Sunchai Payungporn**[4], **David J. Hampson**[5], **Nuvee Prapasarakul**[1,6]*

**1** Department of Veterinary Microbiology, Faculty of Veterinary Science, Chulalongkorn University, Bangkok, Thailand, **2** College of Public Health Sciences, Chulalongkorn University, Bangkok, Thailand, **3** Program in Bioinformatics and Computational Biology, Graduate School, Chulalongkorn University, Bangkok, Thailand, **4** Research Unit of Systems Microbiology, Department of Biochemistry, Faculty of Medicine, Chulalongkorn University, Bangkok, Thailand, **5** School of Veterinary Medicine, Murdoch University, Perth, Western Australia, Australia, **6** Center of Excellence in Diagnosis and Monitoring of Animal Pathogens, Chulalongkorn University, Bangkok, Thailand

* Nuvee.p@chula.ac.th

## Abstract

This study used metagenomic analysis to investigate the gut microbiota and resistome in piglets that were or were not challenged with enterotoxigenic *Escherichia coli* (ETEC) and had or had not received dietary supplementation with microencapsulated probiotics. The 72 piglets belonged to six groups that were either non-ETEC challenged (groups 1–3) or ETEC challenged (receiving 5ml of $10^9$ CFU/ml pathogenic ETEC strain L3.2 one week following weaning at three weeks of age: groups 4–6). On five occasions at 2, 5, 8, 11, and 14 days of piglet age, groups 2 and 5 were supplemented with $10^9$ CFU/ml of multi-strain probiotics (*Lactiplantibacillus plantarum* strains 22F and 25F, and *Pediococcus acidilactici* 72N) while group 4 received $10^9$ CFU/ml of *P. acidilactici* 72N. Group 3 received 300mg/kg chlortetracycline in the weaner diet to mimic commercial conditions. Rectal faecal samples were obtained for metagenomic and resistome analysis at 2 days of age, and at 12 hours and 14 days after the timing of post-weaning challenge with ETEC. The piglets were all euthanized at 42 days of age. The piglets in groups 2 and 5 were enriched with several desirable microbial families, including *Lactobacillaceae*, *Lachnospiraceae* and *Ruminococcaceae*, while piglets in group 3 had increases in members of the *Bacteroidaceae* family and exhibited an increase in *tetW* and *tetQ* genes. Group 5 had less copper and multi-biocide resistance. Mobile genetic elements IncQ1 and IncX4 were the most prevalent replicons in antibiotic-fed piglets. Only groups 6 and 3 had the integrase gene (*intI*) class 2 and 3 detected, respectively. The insertion sequence (IS) 1380 was prevalent in group 3. IS3 and IS30, which are

**Data Availability Statement:** All relevant data are within the paper and its Supporting Information files. The datasets of raw metagenomic sequences were deposited in NCBI Sequence Read Archive (SRA) and are available in the BioProject under the accession number PRJNA769425 (https://www.ncbi.nlm.nih.gov/sra/PRJNA769425).

**Funding:** This study was granted by the 100th Anniversary Chulalongkorn University Fund for Doctoral Scholarship, and the 90th Anniversary Chulalongkorn University Fund (Ratchadaphiseksomphot Endowment Fund) to Prasert Apiwatsiri. In addition, this research also received support from the Agricultural Research Development Agency (ARDA) (Public Organization), Thailand, Pathogen Bank, Faculty of Veterinary Science, Chulalongkorn University, Chulalongkorn Academic Advancement into Its 2nd Century Project, and the CHE-TRF Senior Research Fund (RTA6280013). The funders had no role in study design, data collection and analysis, decision to publish, or preparation of the manuscript.

**Competing interests:** The authors have declared that no competing interests exist.

connected to dietary intake, were overrepresented in group 5. Furthermore, only group 5 showed genes associated with detoxification, with enrichment of genes associated with oxidative stress, glucose metabolism, and amino acid metabolism compared to the other groups. Overall, metagenomic analysis showed that employing a multi-strain probiotic could transform the gut microbiota, reduce the resistome, and boost genes associated with food metabolism.

## Introduction

The gut microbiota of the pig plays a critical role in maintaining health and productivity through supporting optimal nutritional, physiological and immunological functions [1, 2]. Piglets in the weaning transition period are exposed to a variety of stressful factors that may disrupt their newly acquired gut microbiome, resulting in poor growth and health [2]. Infection with enterotoxigenic and verotoxigenic *Escherichia coli* (ETEC and VTEC) are known to cause post-weaning diarrhoea, which results in increased morbidity and mortality, decreased average daily gain (ADG), and the need for increased administration of antibiotics, which all contribute to financial losses for the pig sector [3, 4]. In response, feed additives such as antibiotics, prebiotics, and probiotics have been used to manipulate the piglet gut micro-ecosystem in order to boost growth, improve health status, and prevent diarrhoea after weaning [5].

Antibiotics have been utilized worldwide in the swine industry for many years in order to increase pig productivity while lowering morbidity and mortality [5, 6]. However, administration of in-feed antibiotics impacts both pathogenic and commensal microbes in the gut, leading to decreased alpha-diversity and causing a microbial shift in the animal gut [7]. For example, oxytetracycline treatment may diminish bacterial diversity and richness in the gut microbiota of piglets, moreover subsequent removal of oxytetracycline for 2 weeks does not completely restore bacterial diversity [8]. Several studies have found that pigs exposed to in-feed antibiotics are more likely to develop infections from members of the *Enterobacteriaceae*, *Spirochaetae*, and *Campylobacteraceae* families [6–8].

Antibiotic-treatment of piglets also can increase the diversity and abundance of antibiotic-resistant genes (ARGs) and mobile genetic elements (MGEs) in the porcine gut: these include genes conferring resistance to aminoglycosides, beta-lactams, chloramphenicol, macrolide-lincosamide-streptogramin B (MLS$_B$), sulfonamides, tetracycline, and vancomycin, as well as class 1 integrons and transposons [9]. Antibiotic usage has negative consequences that may affect public health, and, as a result many countries including Thailand have banned the use of antibiotics in livestock agriculture [6]. Consequently, the use of non-antibiotic alternatives for stimulating growth and altering the gut microbiome has received considerable attention in the livestock industries [10].

Probiotics are live microorganisms that are a non-antibiotic option for maintaining gut health, and they have been thoroughly researched over the years [7]. Probiotic supplementation has been shown to have various benefits for humans and animals, including altering the gut microbiota, enhancing food utilization, strengthening gut immunity, and reducing enteric disease [5, 11, 12]. The intestinal microbiota of pigs that were supplemented with *Lactiplantibacillus plantarum* PFM105 was found to be enriched by desirable bacterial families such as *Prevotellaceae* and *Bifidobacteriaceae*, which improve nutrient absorption and have anti-inflammatory activity [7]. Pigs supplemented with 2.5×10$^7$ CFU/ml of *Lactiplantibacillus plantarum* JDFM LP11 showed significantly increased gut microbial richness and diversity, and an

increased *Ruminococcaceae* relative abundance of up to 25% compared to a control group [13]. The effects of probiotics on decreasing the human gut resistome have been studied [14]. For example, infants who received *Bifidobacterium longum* subsp. *infantis* EVC001 had a 90% reduction in ARG abundance when compared to a control group [14]. Unfortunately, to date there have been relatively few comparable studies on the effect of probiotics on modulating the pig gut resistome [15]. Importantly, studies on the pig resistome may provide better insight into antimicrobial resistance (AMR) issues that impact on AMR transmission from pigs to pork consumers.

In our previous studies, several probiotic strains, including *Lactiplantibacillus plantarum* strains 22F and 25F (L22F and L25F) and *Pediococcus acidilactici* strain 72N (P72N), showed excellent safety features, including lack of antimicrobial-resistance genes based on the European Food Safety Authority (EFSA) criteria [16]. Furthermore, they demonstrated promising antibacterial, antiviral, anticonjugation, and antibiofilm action *in vitro* [17–19]. In addition, we previously created a method for preserving our probiotic strains in the form of double-coated microencapsulation for use in pig farms. In an *in vivo* investigation, these probiotic strains used at a final concentration at $10^9$ CFU/ml improved intestinal health and growth development in pigs during the rearing cycle [20, 21]. The purpose of the current study was to undertake whole-metagenome shotgun sequencing on faecal samples to investigate how feeding microencapsulated single-strain and multi-strain probiotics to neonatal pigs influenced their gut microbiota and modulated carriage of ARGs. The study also examined changes in the microbiota that were associated with feeding chlortetracycline or that resulted from ETEC challenge after weaning.

## Materials and methods

### Animals and housing

The experiments performed in this study were approved by the Institutional Animal Care and Use Committee of the Thai Food Research Center, Thai Foods Group (TFG) Public Company Limited (PLC.) under protocol no. 6112–01, and the Feed Research and Innovation Centre, Charoen Pokphand Foods (CPF) Public Company Limited (PLC.) under protocol no. FRI-C-ACUP-1707013. All animal usage and procedures were performed in compliance with the International Guiding Principles for Biomedical Research Involving Animals. The euthanasia procedures were performed following the guidelines for the euthanasia of animals, in compliance with the American Veterinary Medical Association (AVMA). The piglets were rendered unconscious by administering intravenous sodium pentobarbital anaesthesia followed by potassium chloride to induce cardiac arrest and death. The use of all bacterial strains, including lactic acid bacteria (LAB) and ETEC, was approved by the Institutional Biosafety Committee, Chulalongkorn University under Biosafety Use Protocol numbers IBC1831044 and IBC1831045, respectively.

A total of 72 two-day-old healthy neonatal piglets (Large White × Landrace × Duroc) were recruited into the study. The production and health data for 60 of the pigs has been published elsewhere [22]. In the current study an additional 12 piglets were included as a positive control group that were administered with chlortetracycline, with these being reared and handled in an identical fashion to the previously described piglets. The 72 piglets were randomly allocated into six experimental groups with male and female replicate pens per group (6 pigs per pen) at the CPF Feed Research and Innovation Centre. At 21 days of age, piglets in all experimental groups were weaned and transferred to the TFG Research Center. Each experimental group was raised in separate rooms with controlled humidity under an evaporative cooling system at 80%. The environment within the building was temperature-controlled at 32 ± 2°C and

$27 \pm 1°C$ for neonatal and weaned piglets, respectively. All piglets were allowed to independently suck the milk from their sows in the neonatal period. They were allowed *ad libitum* access to a basal diet and water in the weaning period. The ingredient composition and nutrient concentration of the weaner diet is presented in the supplementary data (**S1 Table**).

### Experimental design and sample collection

Information about the treatments received by the six experimental groups is summarized in **Table 1** and **S1 Fig.** The three groups supplemented with probiotics received these on five occasions, when the piglets were 2, 5, 8, 11, and 14 days of age, followed our previous study [20].

Following weaning at 21 days of age, pigs in groups 1–3 were not challenged with ETEC, but received 3 ml of sterile peptone water (Becton, Dickinson and Company, Maryland, USA) at the same time that the ETEC groups (groups 4–6) were challenged. Piglets in the negative control group (group 1) were fed with a basal diet without probiotic and antibiotics. Piglets in the probiotic control group (group 2) were orally supplemented with a 3 ml double-coated multi-strain LAB mixture (L22F, L25F, and P72N) suspended in sterile peptone water at a final concentration at $10^9$ CFU/ml through sterile syringe, receiving this on the five occasions mentioned above. Following weaning, piglets in the antibiotic group (group 3) were fed with a basal diet mixed with antibiotic (chlortetracycline at 300mg/kg), as previously described [20].

In the ETEC challenged groups (groups 4–6), piglets in all groups were fed with a basal diet after weaning. Those in the single strain group (group 4) as neonates previously had been orally supplemented with 3 ml of double-coated single-strain LAB (P72N) suspended in sterile peptone water at a final concentration at $10^9$ CFU/ml via sterile syringe, whilst those in the multi-strain group (group 5) had been orally supplemented with 3 ml of double-coated multi-strain LAB mixture (L22F, L25F, and P72N) suspended in sterile peptone water at a final concentration at $10^9$ CFU/ml through sterile syringe. The piglets in the ETEC control group (group 6) only received 3 ml of sterile peptone water. All piglets in the three ETEC challenged group were orally inoculated with ETEC strain L3.2 at a final concentration at $5\times10^9$ CFU/ml at 28 days of age (7 days after weaning).

Faeces samples were obtained from individual piglets through digital stimulation of the rectum. Approximately five grams of faeces were collected from four of the piglets (2 male and 2 female) in each group on Day 2, 12 hours post-challenge (hpc) and 14 days post-challenge (dpc), with different pigs sampled at each collection. For each group and each collection time, the four faecal samples were combined into one pooled sample before genomic DNA

**Table 1. Summary of the experimental groups.**

| No. | Experimental group | Probiotic supplementation | | | ETEC infection | Antibiotic administration |
|---|---|---|---|---|---|---|
| | | *P. acidilactici* 72N (P72N) | *L. plantarum* 22F (L22F) | *L. plantarum* 25F (L25F) | | |
| **Non-ETEC infection** | | | | | | |
| 1 | Negative control | - | - | - | - | - |
| 2 | Probiotic control | + | + | + | - | - |
| 3 | Antibiotic | - | - | - | - | + |
| **ETEC infection** | | | | | | |
| 4 | Single-strain | + | - | - | + | - |
| 5 | Multi-strain | + | + | + | + | - |
| 6 | ETEC control | - | - | - | + | - |

+ and–indicate with or without probiotic supplementation, antibiotic administration or ETEC infection.

extraction. Faeces were collected into sterile containers and stored at -20˚C until processed within a week of collection.

## DNA extraction and shotgun metagenomic sequencing

Total genomic DNA was extracted from each pooled faecal sample from four piglets per treatment per timepoint using the Quick-DNA/soil microbe microprep kit (Zymoresearch, CA, USA) according to the manufacturer's recommendation. The extracted DNA was checked for purity by $A_{260}/A_{280}$ comparison using the OneDrop TOUCH lite micro-volume spectrophotometer (Biometrics Technologies, Wilmington, DE, USA). DNA degradation was checked by 2% agarose gel electrophoresis (Vivantis, Selangor Darul Ehsan, Malaysia) and visualized under UV in the Syngene™ Ingenius 3 Manual Gel Documentation System (SynGene InGenius, Cambridge, UK). In addition, the total DNA concentration was measured using a Qubit™ 4 fluorometer with the dsDNA broad-range assay kit (Invitrogen™, Thermo Fisher Scientific, Waltham, USA). Shotgun metagenomic sequencing was undertaken using the Illumina Novaseq 6000 on the Illumina HiSeq-PE150 platform at 10-GB data output according to the manufacturer's instructions (Novogene Bioinformatics Technology Co. Ltd., Beijing, China).

## Quality control

The paired-end raw sequence reads were quality filtered in several steps for removing sequencing adapters and low-quality sequences with quality scores <30 using Trimmomatic v.0.36.5 [23]. Finally, any sequences mapped to the pig genome (*Sus scrofa*, NCBI accession no. NC010443) were filtered out using Bowtie2 v.2.3.4.32 [24]. All the bioinformatic analyses were performed on the European Galaxy server (https://usegalaxy.eu/).

## Taxonomic annotation

The taxonomic classifications of the metagenome datasets were identified by Kraken2 (Galaxy Version 2.0.85) (k = 35, ℓ = 31). The Kraken2 database, the complete genomes in RefSeq for the bacterial, archaeal, and viral domains, the human genome and a collection of known vectors were all retrieved from NCBI [25]. Alpha diversity (Species richness, Shanon and Simpson diversity index) and beta-diversity (Bray-Curtis dissimilarity matrix) were analyzed with the QIIME2 platform version 2021.4 (https://qiime2.org/) [26].

## Antibiotic resistance, metal resistance and biocide resistance gene annotation

The clean raw reads after the quality filtering processes were used for similarity searches against the antimicrobial resistance, metal resistance and biocide resistance MEGARes database [27] by using NCBI BLAST+ blastn (Galaxy Version 2.10.1) [28]. The MEGARes database that contains the sequences of approximately 7,868 nucleotide sequences of antimicrobial resistance genes (ARGs) based on a nonredundant compilation of sequences contained in ResFinder, ARG-ANNOT, the Comprehensive Antibiotic Resistance Database (CARD, the National Center for Biotechnology Information (NCBI) Lahey Clinic beta-lactamase archive and BacMet was accessed on 14-10-2019.

## MGEs annotation

The clean raw reads after the quality filtering processes were used for similarity searches for plasmids using the PlasmidFinder database [29] and for class 1, 2, and 3 integron integrase genes in the INTEGRALL database [30, 31] by using NCBI BLAST+ blastn (Galaxy Version

2.10.1) [28]. The PlasmidFinder database contains approximately 469 nucleotide sequences accessed on 13-07-2020, whereas the INTEGRALL database contains 11 nucleotide sequences related to class 1, 2, and 3 integron integrase genes. After the quality filtering processes, the clean raw reads were used for similarity searches against insertion sequences in the ISFinder database [32] by using Diamond (Galaxy Version 0.9.21.0) [33]. The ISFinder database contains approximately 8,836 amino acid sequences and was accessed on 6-10-2020.

Additionally, the confidence match to those databases associated with antibiotic resistance genes and mobile genetic elements was set by considering both percent identity cutoff at 90% and minimum query coverage at 80%, as suggested elsewhere [8, 31]. Moreover, the results of taxonomic profiles, antibiotic resistance, and mobile genetic elements were illustrated in the form of relative abundance by the total count method, which was performed as previously described [34].

### Functional annotation

The clean raw reads from each sample were *de novo* metagenomic assembled with default settings using MEGAHIT (Galaxy Version 1.1.3.43) [35]. The assembled contigs were examined for genome assembly quality using Quast (Galaxy Version 5.0.24) [36]. Functional annotation was determined through metagenome rapid annotation using subsystem technology server version 4 (MG-RAST) [37]. The assembled contigs were submitted to MG-RAST and functional annotation, and they were performed against the Kyoto Encyclopedia of Genes and Genomes database (KEGG) database for analyzing metabolism and SEED subsystem database for analyzing stress response which applied the following thresholds: >60% identity, 15 amino acids for a minimum alignment length, and e-value <1e-5. The investigated markers of stress response were catalase, fumarate and nitrate reduction regulatory protein, iron-binding ferritin-like antioxidant protein, redox-sensitive transcriptional regulator, superoxide dismutase and transcriptional regulator. In addition, the functional results were presented in normalized abundance which was generated by MG-RAST using DESeq analysis, as suggested elsewhere [2].

## Results

### Overall sequencing data and microbial diversity of the piglet faecal samples

DNA extracted from piglet faeces was sequenced with Illumina Hi-seq, obtaining 1.4 billion reads with read counts ranging from 68.9 to 115.2 million. After quality filtering, 1.2 billion high-quality readings were acquired, resulting in an 89.47 percent clean-read rate (S2 Table). After *de novo* metagenomic assembly by MEGAHIT, there were 133,927 to 624,196 assembled contigs (S3 Table). The species richness and diversities (Shanon and Simpson) of gut microbial alpha diversity were lower in the probiotic control group than in the negative control and antibiotic groups within the non-ETEC challenged groups at 12-hours and 14-days after the time of ETEC challenge. However, amongst the ETEC challenged groups, the multi-strain group tended to have greater alpha diversity than the single-strain and ETEC control groups, in terms of both species richness and diversity (S4 Table). The principal coordinate analysis (PCoA) plot on Day 2 (two days of age; before probiotic treatment), at hour 12 (12-hour post-ETEC infection, 12 hpc), and at day 42 (14 days post-ETEC infection, 14 dpc) demonstrated three different clusters, as shown in S2 Fig.

### Taxonomic abundance and composition of the piglet gut microbiota

The abundance and composition of bacterial taxonomic groups at the phylum, family, and genus level are depicted in Fig 1. The most prevalent phyla at 2 days of age (Day 2) were

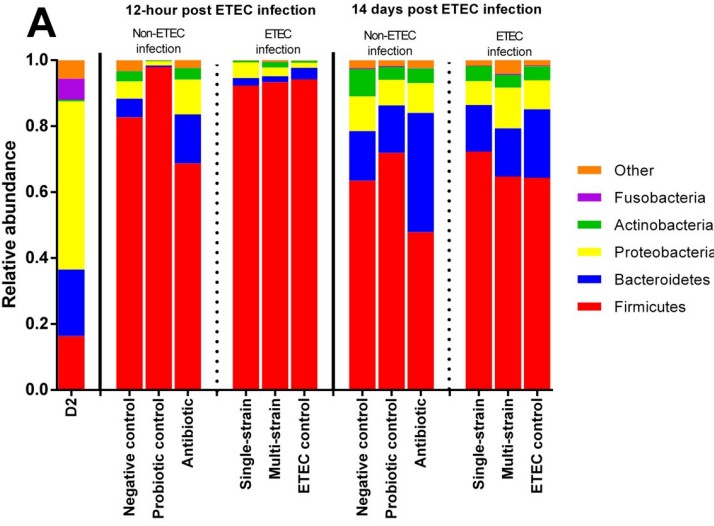

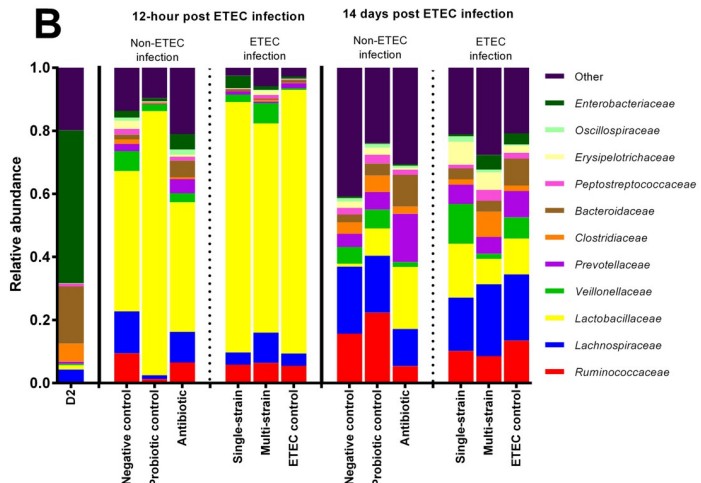

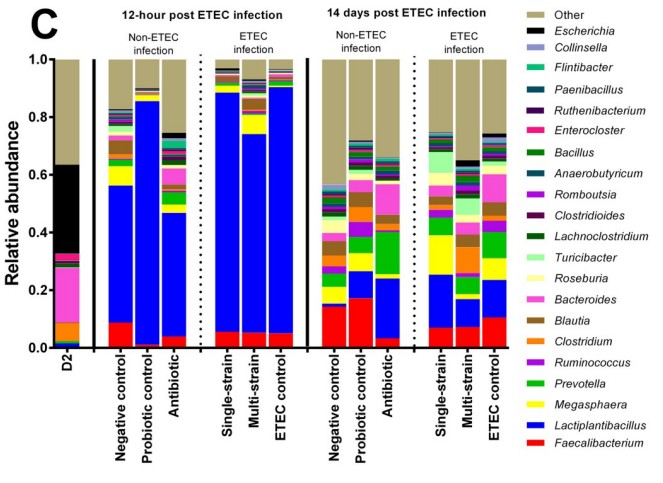

**Fig 1.** The relative abundance of faecal taxonomic classification across treatments in each time-point at the phylum (A), family (B), and genus levels (C), respectively, according to annotation with the Kraken2 database. D2 refers to 2 days of age, before probiotic treatment.

Proteobacteria and Bacteroidetes (**Fig 1A**). The two most common families that were identified were *Enterobacteriaceae* and *Bacteroidaceae* (**Fig 1B**). Furthermore, at Day 2, piglet faeces samples were enriched in the genera *Escherichia* and *Bacteroides* (**Fig 1C**).

The average relative abundance of the Firmicutes, Bacteroidetes, and Proteobacteria phyla was approximately 97% of the total abundance at 12 hpc (**Fig 1A**). In the non-ETEC infected groups, the probiotic control group had a higher proportion of members of the Firmicutes phylum and *Lactobacillaceae* family, while Proteobacteria were found in the highest abundance in the antibiotic group (**Fig 1A and 1B**). The antibiotic group had a higher percentage of *Bacteroidaceae* than the other groups (**Fig 1B**). Furthermore, the probiotic control group had an increased quantity of *Lactiplantibacillus* genus (**Fig 1C**). In the ETEC challenged groups, Firmicutes were found to be the most abundant in all experimental groups, at more than 92% (**Fig 1A**). Firmicutes phylum members *Lachnospiraceae*, *Veillonellaceae* and *Ruminococcaceae* were increased in the multi-strain group (**Fig 1B**). In addition, when compared to the single-strain and ETEC control groups, the relative abundance of *Megasphaera*, *Blautia* and *Ruminococcus* was higher in the multi-strain group (**Fig 1C**).

At 14 dpc, the dominating phyla showed a similar trend as at 12 hpc, with Firmicutes, Bacteroidetes, and Proteobacteria enriched across the experimental groups (**Fig 1A**). In the non-ETEC infected groups, members of the Firmicutes phylum and *Ruminococcaceae* family were found in greater abundance in the probiotic control group than in the other groups, while the *Bacteroidetes* phylum and *Bacteroidaceae* family were still prominent in the antibiotic group (**Fig 1A and 1B**). Furthermore, piglets in the probiotic control group showed higher levels of the genera *Faecalibacterium*, *Megasphaera* and *Ruminococcus* (**Fig 1C**). All the ETEC challenged groups exhibited a high proportion of members of the Firmicutes phylum (**Fig 1A**). At the family level, *Lachnospiraceae* and *Clostridiaceae* were markedly increased in the multi-strain group. In contrast, a high abundance of *Bacteroidaceae* also was observed in the ETEC control group (**Fig 1B**). Furthermore, the genera *Clostridium* and *Bacillus* were enriched in the multi-strain group. At the same time, the ETEC control group had a higher number of *Bacteroides* genus than the other groups (**Fig 1C**).

## Abundance and composition of the piglet gut resistome

At Day 2, TEM genes associated with beta-lactam resistance were the most prominent antimicrobial resistance (AMR) determinants (**Fig 2A and 2B**). The beta-lactam resistance class was enriched in the negative control and antibiotic groups of the non-ETEC infected groups at 12 hpc (**Fig 2A**). In addition, the *tetW* and *tetQ* genes were overrepresented in those groups (**Fig 2B**). Beta-lactam resistance in the ETEC challenged groups was lower in the multi-strain group than in the single-strain and ETEC control groups (**Fig 2A**). Furthermore, the single-strain and ETEC control groups had more TEM and *tetQ* genes than the multi-strain group (**Fig 2B**).

At 14 dpc, amongst the non-ETEC infected groups beta-lactam resistance was dominant in the antibiotic group (**Fig 2A**). In the antibiotic group, the *tetQ*, *mefA* and *tetM* genes were all found in abundance (**Fig 2B**). Furthermore, in the ETEC challenged groups, the *tetQ*, *mefA* and *tetM* genes were less frequent in the multi-strain group than in the single-strain and ETEC control groups (**Fig 2A and 2B**).

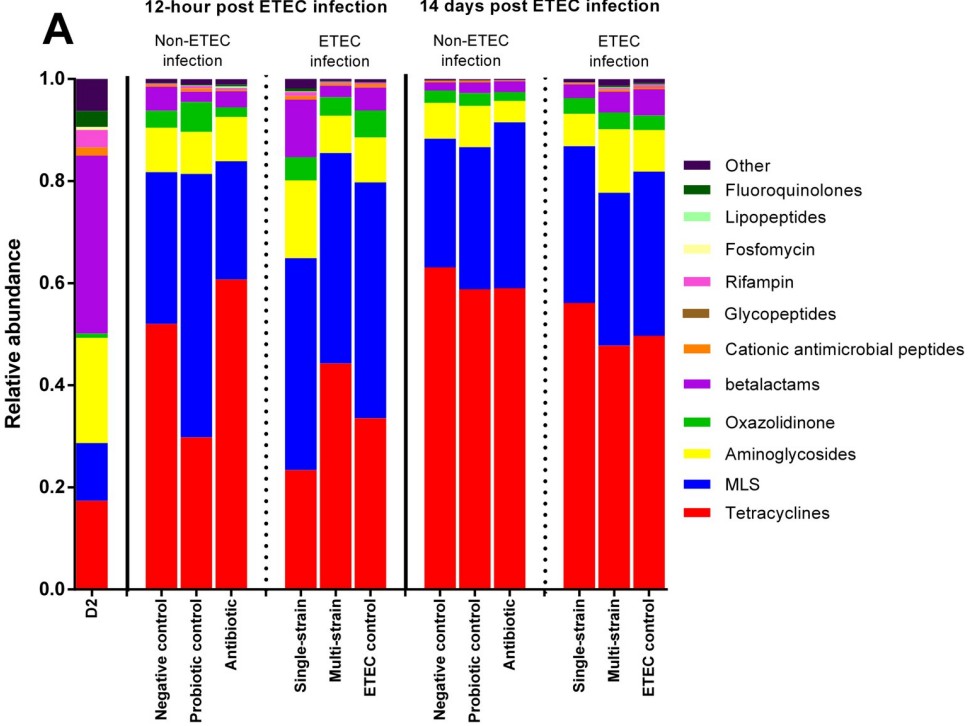

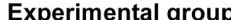

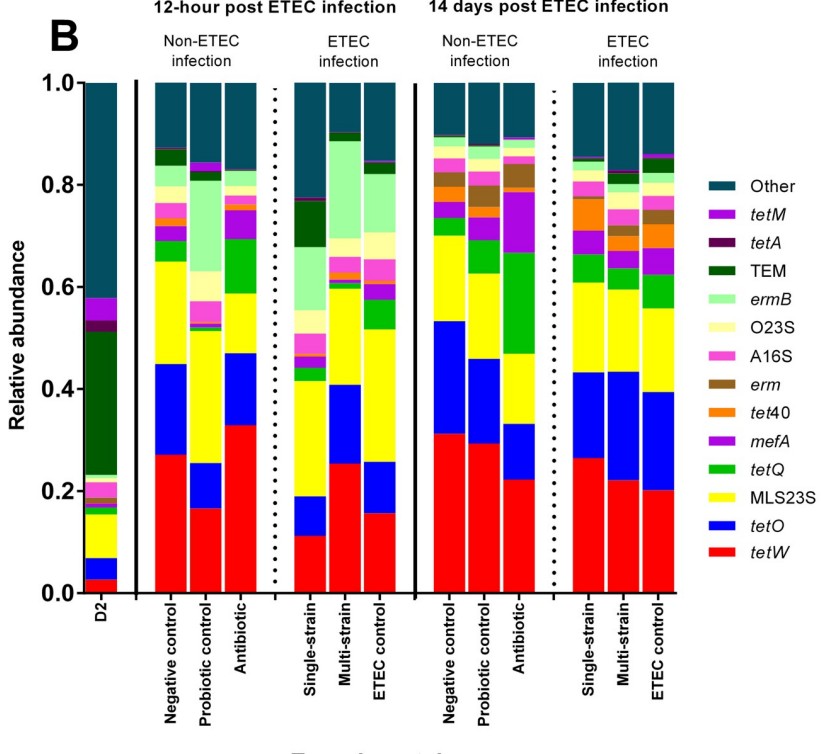

**Fig 2. The relative abundance distribution of faecal antimicrobial resistance classes (A) and groups (B) across treatments at each time-point based on annotation with MEGARes database.** D2 refers to 2 days of age, before probiotic treatment.

## Abundance and diversity of metal and biocide resistance

According to the metal resistance analysis, multi-metal resistance was the most common type identified, followed by copper (Cu) and zinc (Zn) resistance (**Table 2**). At 12 hpc and 14 hpc, the Cu and Zn resistances were more abundant in the antibiotic group than in the negative control and the probiotic control groups. Moreover, Cu resistance in the single-strain group was higher than in the multi-strain and the ETEC control groups (**Table 2**).

Multi-biocide resistance was the most common biocide resistance, followed by acid and acetate resistance. At 12 hpc and 14 hpc, amongst the non-ETEC infected groups the multi-biocide resistance in the probiotic control group was lower than in the negative control and antibiotic groups. The multi-strain group had lower multi-biocide resistance and more abundant peroxide resistance than the single-strain and ETEC control groups in the ETEC infection groups (**Table 3**).

## Mobile genetic elements (plasmid replicons, integron integrase genes and insertion sequences) within the piglet gut microbial community

The antibiotic group had higher levels of several plasmid replicons, including IncQ1, IncX4, IncHI2, and IncHI2A than the other groups (**Fig 3A**). Integrase gene (*intI*) class 1 was the most common integron in all experimental groups, accounting for more than 97% of all detected integrons. Furthermore, an *intI* class 2 was found in the ETEC control group at 14 dpc, whereas an *intI* class 3 was only found in the antibiotic group (**Fig 3B**). At 12 hpc and 14 dpc, insertion sequence (IS) 1380 was enriched in the negative control and antibiotic groups (**Fig 3C**). IS1380 was prominently detected in the ETEC infected groups, while IS3 and IS30 were prominently detected in the single-strain and multi-strain groups (**Fig 3C**).

**Table 2. The percentage of relative abundance of metal resistance group based on metal resistance genes in piglet faecal samples.**

| Metal resistance group | D2 | 12-hours post ETEC challenging | | | | | | 14-days post ETEC challenging | | | | | |
|---|---|---|---|---|---|---|---|---|---|---|---|---|---|
| | | Non-ETEC infection | | | ETEC infection | | | Non-ETEC infection | | | ETEC infection | | |
| | | Negative control | Probiotic control | Antibiotic | Single-strain | Multi-strain | ETEC control | Negative control | Probiotic control | Antibiotic | Single-strain | Multi-strain | ETEC control |
| Multi-metal | 57.11 | 67.27 | 67.50 | 59.07 | 47.83 | 66.15 | 55.56 | 55.62 | 61.78 | 57.14 | 60.77 | 56.79 | 55.88 |
| Copper | 13.31 | 10.12 | 8.93 | 14.65 | 35.75 | 10.76 | 12.59 | 11.23 | 15.53 | 42.86 | 12.10 | 12.00 | 19.80 |
| Nickel | 8.96 | 7.00 | 7.02 | 8.22 | 5.37 | 6.31 | 9.35 | 7.40 | 7.05 | 0.00 | 9.86 | 9.24 | 7.98 |
| Zinc | 8.52 | 6.38 | 6.84 | 7.77 | 4.17 | 7.17 | 9.45 | 12.05 | 7.38 | 0.00 | 6.57 | 8.42 | 7.54 |
| Arsenic | 5.41 | 5.65 | 4.91 | 5.36 | 3.36 | 5.42 | 5.59 | 6.58 | 4.96 | 0.00 | 6.64 | 6.89 | 4.90 |
| Sodium | 4.93 | 3.15 | 3.80 | 4.22 | 2.98 | 3.63 | 5.25 | 1.92 | 2.64 | 0.00 | 3.08 | 5.12 | 3.27 |
| Iron | 0.66 | 0.02 | 0.23 | 0.03 | 0.01 | 0.12 | 0.02 | 1.64 | 0.00 | 0.00 | 0.00 | 0.00 | 0.00 |
| Chromium | 0.62 | 0.41 | 0.54 | 0.43 | 0.50 | 0.41 | 0.64 | 1.10 | 0.00 | 0.00 | 0.56 | 0.62 | 0.49 |
| Mercury | 0.46 | 0.00 | 0.20 | 0.00 | 0.01 | 0.03 | 0.28 | 2.47 | 0.66 | 0.00 | 0.00 | 0.00 | 0.15 |
| Tellurium | 0.01 | 0.00 | 0.03 | 0.25 | 0.01 | 0.00 | 1.27 | 0.00 | 0.00 | 0.00 | 0.42 | 0.91 | 0.00 |

D2 refers to 2 days of age, before probiotic treatment.

**Table 3. The percentage of relative abundance of metal resistance group based on biocide resistance genes in piglet faecal samples.**

| Biocide resistance group | D2 | 12-hours post ETEC challenging | | | | | | 14-days post ETEC challenging | | | | | |
|---|---|---|---|---|---|---|---|---|---|---|---|---|---|
| | | Non-ETEC infection | | | ETEC infection | | | Non-ETEC infection | | | ETEC infection | | |
| | | Negative control | Probiotic control | Antibiotic | Single-strain | Multi-strain | ETEC control | Negative control | Probiotic control | Antibiotic | Single-strain | Multi-strain | ETEC control |
| Acid | 34.427 | 32.567 | 29.961 | 37.233 | 31.562 | 32.426 | 32.642 | 27.723 | 35.223 | 0 | 29.978 | 30.665 | 30.196 |
| Multi-biocide | 34.392 | 34.602 | 26.800 | 32.058 | 37.785 | 33.557 | 36.889 | 38.614 | 27.935 | 100 | 40.940 | 36.569 | 37.444 |
| Acetate | 18.453 | 18.394 | 20.185 | 14.820 | 16.721 | 17.470 | 19.358 | 27.723 | 12.955 | 0 | 16.555 | 21.628 | 18.806 |
| Peroxide | 8.905 | 10.969 | 9.922 | 11.755 | 10.628 | 13.322 | 7.556 | 3.960 | 21.053 | 0 | 11.186 | 7.948 | 10.655 |
| Phenolic compound | 3.799 | 3.468 | 3.210 | 4.134 | 3.304 | 3.226 | 3.556 | 1.980 | 2.834 | 0 | 1.342 | 3.191 | 2.899 |
| Biguanide | 0.007 | 0 | 0 | 0 | 0 | 0 | 0 | 0 | 0 | 0 | 0 | 0 | 0 |
| Quaternary ammonium compounds | 0.016 | 0 | 9.922 | 0 | 0 | 0 | 0 | 0 | 0 | 0 | 0 | 0 | 0 |
| Paraquat | 0.001 | 0 | 0 | 0 | 0 | 0 | 0 | 0 | 0 | 0 | 0 | 0 | 0 |

D2 refers to 2 days of age, before probiotic treatment.

## Microbial functional diversity of the gut metagenome related to stress response in ETEC and non-ETEC infected piglets

The results for the stress response that was analyzed using the SEED subsystem database within the MG-RAST server are summarised in **S3 Fig** and **Fig 4**. In all experimental groups, oxidative stress was the most prevalent response, ranging from 33.20 to 45.63% in the stress response at level 2. Surprisingly, the multi-strain group had the highest stress response associated with detoxification at 12 hpc, accounting for more than 19% of the total (**S3 Fig**). The transcriptional and redox-sensitive transcriptional regulators, which were the main markers of oxidative stress responses in this study, were found in approximately 80% of the total sequences in the probiotic control and multi-strain groups. In addition, compared to the other groups, the multi-strain group had more catalase and superoxide dismutase (**Fig 4**).

## Microbial functional diversity of the gut metagenome associated with nutrient metabolism in ETEC and non-ETEC infected piglets

The relative abundance of functional genes at level 1 KEGG related to metabolism is shown in **S4 Fig**. Amino acid and carbohydrate metabolism were dominant in roughly 60% of the total nutrient metabolism sequences (**S4 Fig**). Most amino acid metabolism pathways involved alanine, aspartate, and glutamate metabolism, followed by glycine, serine and threonine metabolism, and cysteine and methionine metabolism (**Table 4**). Furthermore, glycolysis/gluconeogenesis, amino sugar and nucleotide sugar metabolism, and galactose metabolism were the top three carbohydrate metabolisms, respectively (**Table 5**). Among the non-ETEC infected groups, amino acid and carbohydrate metabolism pathways were less represented in the probiotic control group than in the antibiotic group at 12 hpc. The multi-strain group, on the other hand, had more genes related to amino acid metabolism than did the other ETEC infected groups (**Table 4**). The probiotic control and multi-strain groups had more genes associated with amino acid and carbohydrate metabolism at 14 dpc than the other groups (**Tables 3 and 4**).

## Discussion

In our previous study that used 60 of the pigs included in the current study, dosing the neonatal piglets with the multi-strain probiotic enhanced average daily gain and feed conversion

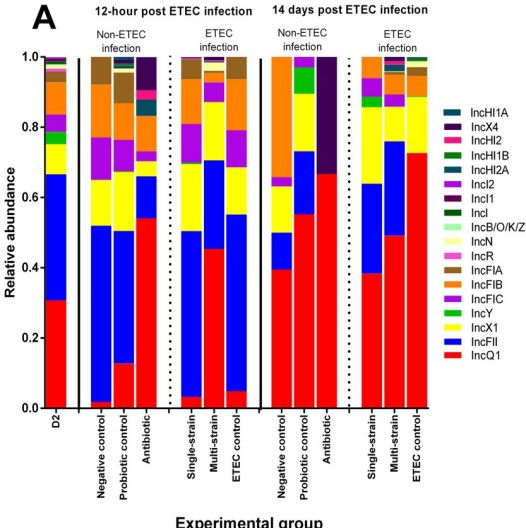

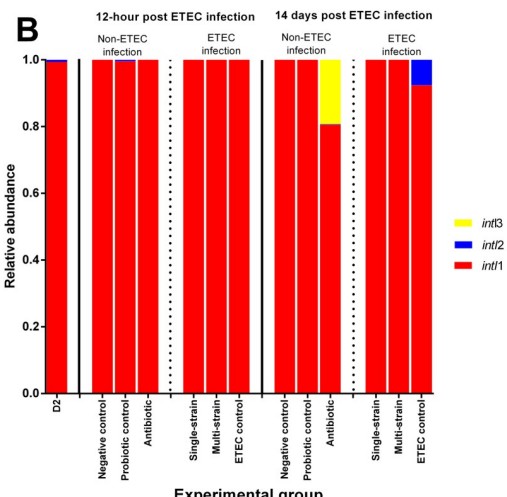

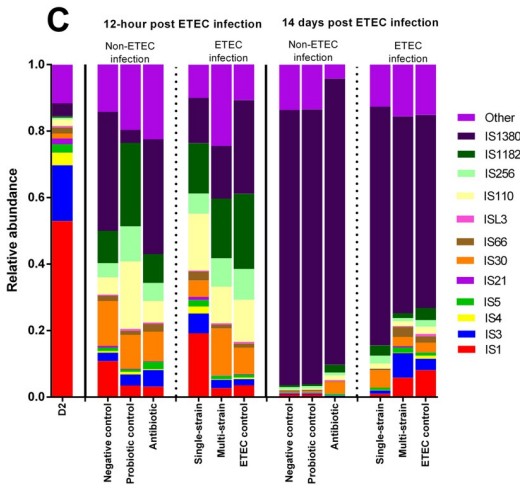

**Fig 3. Relative abundance of genes based on mobile genetic element annotation across treatments in each time-point.** Stacked bar plot demonstrating the relative abundance distribution of classified plasmid replicons (A). Stacked bar plot illustrating the relative abundance distribution of aligned integron integrase genes (B). Stacked bar plot displaying the relative abundance distribution of sorted insertion sequences (C). D2 refers to 2 days of age, before probiotic treatment.

ratio (FCR) of the piglets after ETEC challenge following weaning, whilst supplementing with the single-strain probiotic increased FCR [22]. The piglets receiving probiotics had an increase in lactic acid bacteria counts and a decrease in *E. coli* counts in the faeces, with lower levels of virulence genes being detected. Challenged piglets receiving probiotics had milder intestinal lesions with better morphology, including greater villous heights and villous height per crypt depth ratios, than pigs just receiving ETEC. This study demonstrated that prophylactic administration of microencapsulated probiotic strains may improve outcomes in weaned pigs with

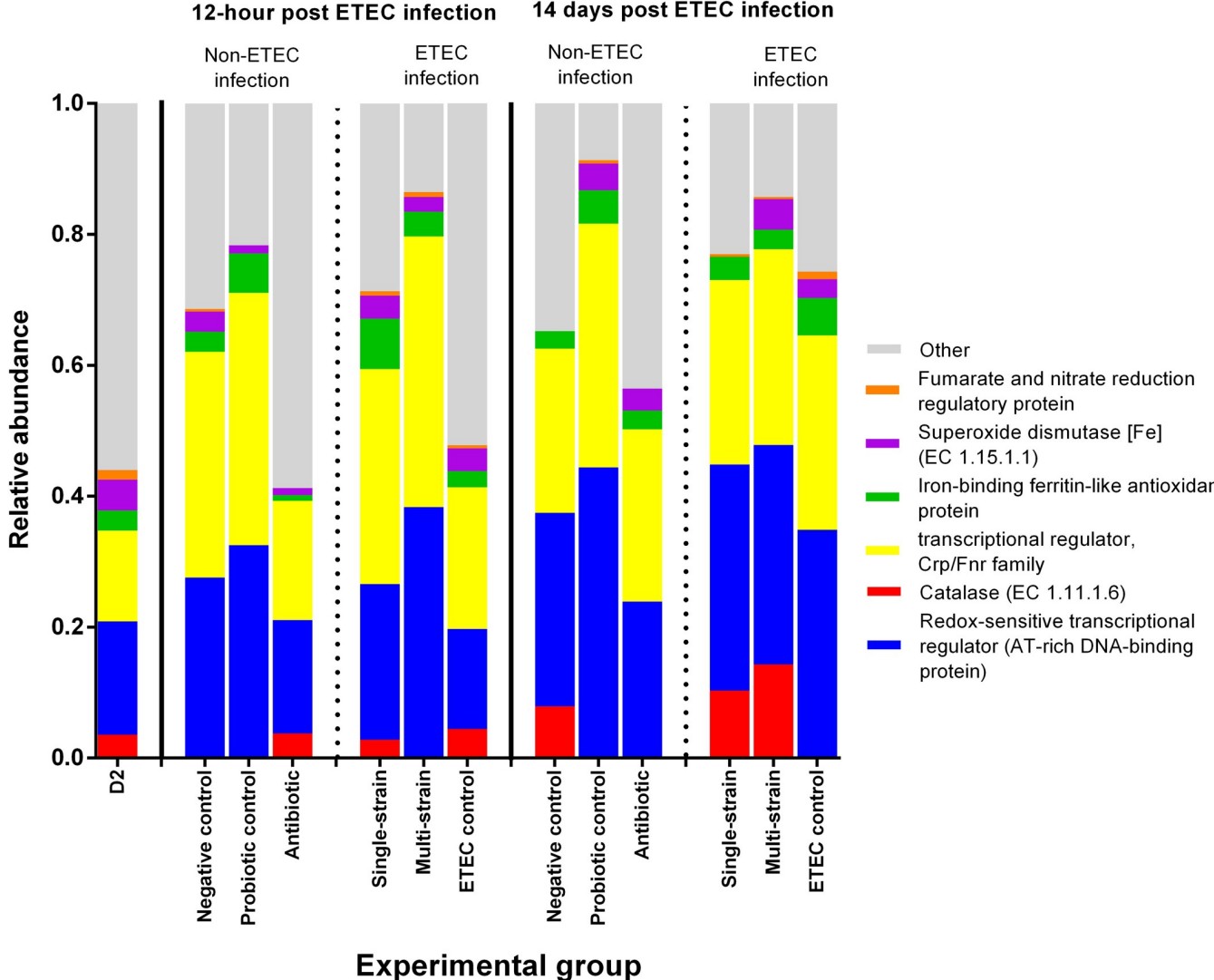

**Fig 4. Relative abundance of the level 4 SEED subsystem classified reads associated with oxidative stress from piglet faecal samples in ETEC or non-ETEC infected piglets.** D2 refers to 2 days of age, before probiotic treatment.

**Table 4. Normalized abundance of the level 3 KEGG functional reads related to amino acid metabolism from faecal samples in ETEC or non-ETEC challenging piglets.**

| Amino acid metabolism | D2 | 12-hours post ETEC challenging | | | | | | 14-days post ETEC challenging | | | | | |
| --- | --- | --- | --- | --- | --- | --- | --- | --- | --- | --- | --- | --- | --- |
| | | Non-ETEC infection | | | ETEC infection | | | Non-ETEC infection | | | ETEC infection | | |
| | | Negative control | Probiotic control | Antibiotic | Single-strain | Multi-strain | ETEC control | Negative control | Probiotic control | Antibiotic | Single-strain | Multi-strain | ETEC control |
| Glycine, serine, and threonine metabolism | 1310 | 1144 | 3799 | 5668 | 2054 | 2732 | 2004 | 3382 | 3715 | 2852 | 4428 | 4668 | 2511 |
| Alanine, aspartate, and glutamate metabolism | 1390 | 1576 | 4677 | 6664 | 2775 | 3241 | 2511 | 4107 | 4512 | 3936 | 5647 | 5694 | 2004 |
| Arginine and proline metabolism | 837 | 792 | 2405 | 3453 | 1377 | 1767 | 1239 | 2213 | 2369 | 1914 | 2751 | 2948 | 1918 |
| Cysteine and methionine metabolism | 825 | 1018 | 3487 | 4642 | 1933 | 2521 | 1918 | 3135 | 3188 | 2629 | 3843 | 4213 | 1239 |
| Lysine biosynthesis | 639 | 740 | 2136 | 2937 | 1222 | 1505 | 1131 | 2011 | 2160 | 1573 | 2557 | 2761 | 1131 |
| Phenylalanine, tyrosine, and tryptophan biosynthesis | 504 | 515 | 1648 | 2320 | 1014 | 1129 | 909 | 1549 | 1667 | 1241 | 2029 | 1975 | 966 |
| Valine, leucine, and isoleucine biosynthesis | 494 | 444 | 1446 | 2191 | 965 | 1108 | 966 | 1515 | 1596 | 1268 | 1779 | 2018 | 919 |
| Histidine metabolism | 501 | 542 | 1493 | 2222 | 847 | 1040 | 919 | 1435 | 1595 | 1226 | 1835 | 1914 | 909 |
| Valine, leucine, and isoleucine degradation | 328 | 194 | 626 | 1146 | 334 | 460 | 314 | 569 | 648 | 525 | 775 | 813 | 314 |
| Phenylalanine metabolism | 300 | 109 | 430 | 566 | 193 | 303 | 173 | 375 | 424 | 249 | 407 | 497 | 173 |
| Tyrosine metabolism | 194 | 157 | 352 | 427 | 186 | 284 | 163 | 341 | 345 | 209 | 401 | 469 | 163 |
| Lysine degradation | 129 | 36 | 121 | 191 | 54 | 94 | 42 | 94 | 127 | 93 | 129 | 160 | 42 |
| Tryptophan metabolism | 41 | 23 | 50 | 95 | 14 | 41 | 21 | 35 | 38 | 40 | 57 | 72 | 21 |

D2 refers to 2 days of age, before probiotic treatment.

colibacillosis. The current study enlarged on these findings by examining the gut microbiota of these pigs in more detail. An additional group of pigs receiving chlortetracycline after weaning was included to help compare probiotics and antimicrobials in influencing the gut microbiota and supporting pig health after weaning. Tetracyclines are commonly given to piglets after weaning to help prevent the development of post-weaning diarrhoea. Whole-genome shotgun metagenomic sequencing of DNA extracted from faeces was used to investigate the gut microbiome, resistome, stress responses, and nutrient metabolism, and to examine how the probiotics cause beneficial changes in piglets infected with ETEC.

Faecal samples were used as a proxy for intestinal samples for examining the gut microbiota, as faeces can be obtained from live pigs which then can be sampled again at later stages. The gut microbiota composition in faeces collected from the rectum seems to be stable, and it shows the same pattern as the hindgut regions, indicating that the faecal microbiota can be used as a proxy for the microbiota in the large intestine of the pigs [38, 39]. Samples were pooled because it was not technically or financially possible to examine samples from all

**Table 5. Normalized abundance of the level 3 KEGG functional reads associated with carbohydrate metabolism from faecal samples in ETEC or non-ETEC infected piglets.**

| Carbohydrate metabolism | D2 | 12-hours post ETEC challenging | | | | | | 14-days post ETEC challenging | | | | | |
|---|---|---|---|---|---|---|---|---|---|---|---|---|---|
| | | Non-ETEC infection | | | ETEC -infection | | | Non-ETEC infection | | | ETEC infection | | |
| | | Negative control | Probiotic control | Antibiotic | Single-strain | Multi-strain | ETEC control | Negative control | Probiotic control | Antibiotic | Single-strain | Multi-strain | ETEC control |
| Glycolysis / Gluconeogenesis | 880 | 830 | 2373 | 3202 | 1250 | 1652 | 1209 | 2194 | 2506 | 1609 | 2773 | 2980 | 2094 |
| Pyruvate metabolism | 848 | 692 | 1801 | 2463 | 975 | 1243 | 977 | 1757 | 1937 | 1276 | 2201 | 2218 | 1655 |
| Amino and nucleotide sugar metabolism | 712 | 684 | 2274 | 3037 | 1245 | 1464 | 1136 | 1961 | 2213 | 1413 | 2593 | 2731 | 1994 |
| Galactose metabolism | 748 | 784 | 2134 | 2668 | 1329 | 1576 | 1171 | 1878 | 2057 | 1627 | 2384 | 2631 | 2006 |
| Pentose phosphate pathway | 703 | 632 | 1769 | 2377 | 923 | 1338 | 866 | 1602 | 1858 | 1162 | 2133 | 2152 | 1563 |
| Starch and sucrose metabolism | 569 | 673 | 1892 | 2653 | 1088 | 1493 | 1061 | 1679 | 1836 | 1458 | 2259 | 2382 | 1764 |
| Fructose and mannose metabolism | 567 | 535 | 1457 | 2360 | 894 | 966 | 820 | 1201 | 1337 | 1204 | 1742 | 1785 | 1289 |
| Pentose and glucuronate interconversions | 554 | 515 | 1392 | 1948 | 863 | 990 | 784 | 1048 | 1424 | 1182 | 1548 | 1507 | 1295 |
| Citrate cycle (TCA cycle) | 318 | 330 | 1146 | 1805 | 562 | 729 | 520 | 1031 | 1065 | 941 | 1362 | 1408 | 888 |
| Glyoxylate and dicarboxylate metabolism | 224 | 158 | 592 | 724 | 318 | 416 | 253 | 616 | 643 | 380 | 641 | 648 | 473 |
| Ascorbate and aldarate metabolism | 92 | 30 | 62 | 68 | 45 | 52 | 34 | 62 | 63 | 59 | 46 | 72 | 66 |
| Butanoate metabolism | 80 | 74 | 256 | 347 | 104 | 188 | 120 | 206 | 238 | 188 | 270 | 285 | 204 |
| Inositol phosphate metabolism | 60 | 28 | 103 | 106 | 38 | 77 | 43 | 82 | 90 | 42 | 90 | 117 | 71 |
| Propanoate metabolism | 58 | 21 | 69 | 80 | 35 | 36 | 36 | 43 | 63 | 34 | 75 | 65 | 44 |
| C5-Branched dibasic acid metabolism | 12 | 0 | 21 | 31 | 0 | 25 | 3 | 26 | 34 | 11 | 18 | 37 | 15 |

D2 refers to 2 days of age, before probiotic treatment.

individual piglets in this study. It is acknowledged that this does not allow comparison of variations between pigs within a group, but this approach was necessary for practical purposes and does provide an overview of group affects. The methodology used means that it was not appropriate to undertake statistical analysis between groups in this study.

Faecal microbial diversity expanded over time during the weaning period, which was consistent with previous findings [8, 40]. Firmicutes and Proteobacteria were the most prevalent phyla found in piglet faeces at Day 2, which agrees with a previous study which found that these phyla were the most prevalent microbial components in early newborn piglets [41]. In addition, the genus *Escherichia*, which belongs to the *Enterobacteriaceae* family, was found in abundance. Pathogenic strains of *Escherichia coli* can have an impact on human and animal health by acquiring and disseminating AMR and virulence genes through the food supply chain, and they act as a biomarker for piglets that may develop diarrhoea in the lactation phase [4, 42].

According to several studies, Firmicutes and Bacteroidetes are the most numerous phyla in the piglet faecal microbiota during the post-weaning phase [4, 8, 13, 40]. The probiotic control group had a larger proportion of the Firmicutes phylum than the other groups in the current study. This result appears to be congruent with another study, which found that supplementing with *Enterococcus faecalis* UC-100 was associated with more than 85% of the total sequences enriched by the Firmicutes phylum [40]. In the current study the genera *Blautia*, *Lactiplantibacillus*, *Megasphaera*, *Ruminococcus*, *Clostridium* and *Faecalibacterium* were identified in the multi-strain and probiotic control groups. These genera are regarded as being beneficial microbes due to a variety of characteristics, including the ability to produce antibacterial substances (e.g., bacteriocins, organic acids) that inhibit growth of pathogens, the ability to increase carbohydrate metabolism by utilizing dietary starch and fiber, and the ability to produce short-chain fatty acids (SCFAs) that reduce gut inflammation [2, 7, 13, 40, 43–46]. The group receiving chlortetracycline showed an increase in Proteobacteria and Bacteroidetes, which is consistent with prior research demonstrating that antibiotic administration could boost these phyla [6–8, 47]. However, several studies have suggested that enhanced numbers of Bacteroidetes may promote host health by enhancing nutrient digestion and absorption [2, 7]. Furthermore, it has been suggested that they act as a biomarker for gut dysbiosis in piglets given antibiotics [48].

We found a variety of AMR determinants in neonatal piglets in this study, and the dominant antibiotic-resistant classes and genes discovered in this study appear to be linked to our previous research, which found that neonatal piglets in antibiotic-free farms had high levels of beta-lactam resistance and carriage of the *blaTEM* gene [49]. The World Health Organization classifies beta-lactams as critically important antimicrobials, meaning they have the potential to have a major impact on human health [50].

Previous studies have shown that tetracyclines and MLSs are the most common antibiotic resistance classes in weaned pigs receiving or not receiving in-feed antibiotics, and the findings of the current study are consistent with this [8, 47]. High levels of beta-lactam resistance also were found in both the negative control and the antibiotic groups. This matches previous findings of dominant beta-lactam resistance in medicated and unmedicated piglets [8]. The antibiotic group had more *tetW*, *tetQ*, *tetM* and *mefA* genes, which are involved in tetracycline ribosomal protection proteins and MLS efflux pumps, on an AMR gene level [8, 51]. These genes have been found on mobile genetic elements such as conjugative transposons, which can spread to other bacteria via horizontal transfer. Furthermore, previous research has found that the *Bacteroidaceae* family frequently carry such genes, suggesting that they could be a source of AMR genes for the gut microbial community [8, 52].

Piglets given probiotics in the current study had a lower proportion of AMR determinants like beta-lactam resistance, *mefA*, *tetQ*, and *tetW* genes than piglets given antibiotics. Probiotics may modify the gut microbial population by reducing the abundance of some antibiotic-resistant microorganisms through a variety of processes, including competition for food substrates and binding sites, production of antimicrobial compounds, and regulation of immune responses [12]. These data are consistent with prior research showing that probiotic treatment in infants can reduce ARG abundance by eliminating antibiotic-resistant carriers [14]. To our knowledge, this is the first study to investigate the effects of probiotic supplementation on modulation of the pig gut resistome. However, since the existence of some antibiotic genes may not indicate phenotypic resistance, a weakness in the current study was the lack of comparison between AMR genotypic and phenotypic features. Phenotypic determinations should be performed on fresh faecal samples, and this was not possible with the frozen samples [6, 47].

Based on co-selection processes such as co-resistance, cross-resistance, and biofilm formation, there is evidence of a link between antibiotic, metal, and biocide resistances [53]. Copper and multi-biocide resistances were found in abundance in the antibiotic group, which was linked to numerous antimicrobial drug resistances such as to beta-lactams, fluoroquinolones, macrolides and tetracyclines [54–56]. This could explain why the antibiotic group had more Cu and multi-biocide resistance than the other groups. Biofilm production is critical for preserving metal and biocide resistances, protecting the population from metal and biocide toxicity, and increasing the lateral transfer of ARGs with co-selected metal resistant genes [53, 55]. Our probiotic strains have been shown to minimize ARG transfer and biofilm development *in vitro* [17]. Taken together, this could be another reason why the probiotic supplemented groups had lower Cu and multi-biocide resistance genes detected.

The complete set of MGEs, and specifically the mobilome, are thought to hasten the spread of ARGs among members of the gut microbiota [57]. In the antibiotic group, IncQ1, IncX4, IncHI2, and IncHI2A plasmids were detected in abundance, which is of concern because it may allow multidrug resistance (MDR) in humans and animals, such as resistance to aminoglycosides, beta-lactams, and tetracycline [58]. Furthermore, they may be involved in colistin resistance where they contain the mobilized colistin resistance (*mcr*) gene [58, 59]. Interestingly, the probiotic-supplemented groups had fewer plasmid replicons than the antibiotic-supplemented group. This finding supports the theory that probiotics can regulate the gut microbial community by lowering the proportion of microbiota carrying certain plasmids, or by blocking ARG transfer via a variety of pathways [12, 17]. In the current study, class 1 integrons were shown to be abundant in all groups. This finding is consistent with prior research that found it to be the most common integron type, accounting for about 80% of all types in enteric bacteria in humans and animals [60]. The ETEC control group contained class 2 integron, which is involved in resistance to aminoglycosides, beta-lactams, and erythromycins [60, 61]. In addition, class 3 integron was found only in the antibiotic group, and it has been linked to beta-lactam resistance and the IncQ plasmid replicon [60]. Furthermore, the antibiotic group had higher levels of IS1380, which can increase beta-lactam and nitroimidazole resistance in Bacteroidetes, the antibiotic group's predominant member [62]. In our study, the probiotic supplemented groups had more IS3 and IS30, which are involve with numerous metabolic modulations such arginine production and the use of acetate, citrate, and galactose [62]. This appears to be the first report to detail the effects of probiotic supplementation on MGE regulation in the pig gut microbial population.

An imbalance between reactive oxygen species (ROS) and antioxidant responses was typically seen in the weaning transition or after ETEC infection, which events are likely to be a source of oxidative stress [2]. Excessive exposure to ROS can have negative consequences on bacterial cells, resulting in protein activity dysfunction and bacterial cell death [2]. In the probiotic groups, genes related to the oxidative response, particularly "transcriptional regulator" and "redox-sensitive transcriptional regulator" contributing to antioxidant activity, were elevated [63]. Furthermore, antioxidant capacity was related to detoxification in the multi-strain group following ETEC challenge [63]. This finding agrees with previous studies suggesting that a variety of probiotic isolates may boost antioxidant defense mechanisms and reduce oxidative stress [5, 64]. Consequently, further research on the antioxidant activities of our probiotic strains (L22F, L25F, and P72N) is needed to improve understanding of the mechanism of stress response modulation.

The probiotic groups had increased numbers of amino acid metabolism genes, which agrees with previous work showing that many bacterial species, including *Bifidobacterium*, *Lactiplantibacillus*, *Megasphaera* and *Veillonella* are involved in modulating amino acid metabolism [5, 65]. Moreover, several amino acids, including alanine, arginine, cysteine, glutamine, glycine,

lysine, methionine and threonine have been shown to benefit pig gut health, including by altering the gut microbiota, maintaining intestinal shape, and increasing gut immunological, anti-inflammatory, and anti-oxidative stress functions [66]. We also found that the probiotic supplemented groups had higher levels of genes involved in glucose metabolism. This result is consistent with previous studies that identified carbohydrate utilization via fermentation and hydrolysis pathways was found in a variety of gut bacteria, including *Bifidobacterium*, *Faecalibacterium*, *Lactiplantibacillus* and *Ruminococcus*, [5, 65]. SCFAs, which are readily available energy sources for pigs, are one of the bacterial metabolites produced following food digestion that may have anti-inflammatory and antagonistic properties [13, 65]. However, additional investigations into the complete genomes of our probiotic strains are recommended to expand these findings. These data should be linked to global metabolomic and proteomic studies to better understand the mechanisms of the probiotic effects on the gut microbiome and resistome.

## Conclusion

In conclusion, supplementing neonatal piglets with our microencapsulated probiotics helped to improve the composition of the gut microbiota by increasing the numbers and proportions of beneficial bacteria. These probiotic effects continued after weaning and were associated with improved performance and an enhanced antioxidant response in piglets challenged with ETEC. The changes in the microbiota benefited the piglets in other ways, including by reducing the antibiotic resistome, metal resistance, biocide resistance, and number of MGEs. Additionally, by enriching genes associated with amino acid and carbohydrate metabolism, probiotics boosted the antioxidant response to reduce oxidative stress and promote improved nutritional utilization. Taken together, these data shed light on probiotic effects on the gut microbiome and on resistome regulation.

## Supporting information

**S1 Fig. Schematic of experimental design and sample collection.** D indicates day after birth and hpc refers to hours post ETEC challenge.
(DOCX)

**S2 Fig. Principal coordinate analysis (PCoA) plot based on Bray-Curtis dissimilarity index of microbial taxonomic profile at the species level from piglet faecal samples across treatments in each time-point.** The geometric shapes demonstrate the group of samples in each time-point. D2 refers to 2 days of age, before probiotic treatment.
(DOCX)

**S3 Fig. Relative abundance of the level 2 SEED subsystem aligned genes associated with stress response from piglet faecal samples in ETEC or non-ETEC infected piglets.** D2 refers to 2 days of age, before probiotic treatment.
(DOCX)

**S4 Fig. The relative abundance of the level 1 KEGG functional genes related to metabolism from piglet faecal samples in ETEC or non-ETEC infected piglets.** D2 refers to 2 days of age, before probiotic treatment.
(DOCX)

**S1 Table. Ingredient composition and nutrient concentration of the experimental basal diet.**
(DOCX)

**S2 Table. Summary of overall sequencing data.**
(DOCX)

**S3 Table. The summary of all *de novo* assembled metagenomic sequence data by using MEGAHIT and determining by QUAST.**
(DOCX)

**S4 Table. Alpha diversity of gut microbial communities from piglet fecal samples across treatments and each time-point.**
(DOCX)

## Acknowledgments

We gratefully thank the Thai Foods Group (TFG) Public Company Limited (PLC.), and the Feed Research and Innovation Center, Charoen Pokphand Foods (CPF) Public Company Limited (PLC.), for affording animals and facilities and assisting with animal care.

## Author Contributions

**Conceptualization:** Prasert Apiwatsiri, Wandee Sirichokchatchawan, Sunchai Payungporn, Nuvee Prapasarakul.

**Data curation:** Prasert Apiwatsiri, Vorthon Sawaswong, Pattaraporn Nimsamer, Sunchai Payungporn.

**Formal analysis:** Prasert Apiwatsiri, Nuvee Prapasarakul.

**Funding acquisition:** Nuvee Prapasarakul.

**Investigation:** Prasert Apiwatsiri, Pawiya Pupa, Vorthon Sawaswong, Nuvee Prapasarakul.

**Methodology:** Prasert Apiwatsiri, Pawiya Pupa, Wandee Sirichokchatchawan, Vorthon Sawaswong, Pattaraporn Nimsamer, Sunchai Payungporn, Nuvee Prapasarakul.

**Resources:** Pawiya Pupa, Wandee Sirichokchatchawan, Pattaraporn Nimsamer.

**Software:** Pattaraporn Nimsamer, Sunchai Payungporn.

**Supervision:** Sunchai Payungporn.

**Validation:** David J. Hampson.

**Visualization:** Prasert Apiwatsiri.

**Writing – original draft:** Prasert Apiwatsiri, Nuvee Prapasarakul.

**Writing – review & editing:** David J. Hampson, Nuvee Prapasarakul.

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
