## [Decision Letter · Decision Letter 0]

8 Apr 2022

PONE-D-22-04406Metagenomic analysis of the gut microbiota in piglets reveals beneficial effects of probiotics on its composition, resistome, digestive function and oxidative stress responsesPLOS ONE

Dear Dr. Prapasarakul,

Thank you for submitting your manuscript to PLOS ONE. After careful consideration, we feel that it has merit but does not fully meet PLOS ONE’s publication criteria as it currently stands. Therefore, we invite you to submit a revised version of the manuscript that addresses the points raised during the review process. Also, please provide a point-by-point document addressing all the concerns of the reviewer.

We look forward to receiving your revised manuscript.

Kind regards,

Horacio Bach

Academic Editor

PLOS ONE

Journal Requirements:

2. In your Methods section, please include a comment about the state of the animals following this research. Were they euthanized or housed for use in further research? If any animals were sacrificed by the authors, please include the method of euthanasia and describe any efforts that were undertaken to reduce animal suffering.

“This research was also supported by Pathogen Bank Project, Chulalongkorn Academic Advancement into Its 2nd Century Project and the CHE-TRF Senior Research Fund (RTA6280013).”

Reviewers' comments:

Reviewer's Responses to Questions

**Comments to the Author**

1. Is the manuscript technically sound, and do the data support the conclusions?

Reviewer #1: Partly

2. Has the statistical analysis been performed appropriately and rigorously? 

Reviewer #1: No

3. Have the authors made all data underlying the findings in their manuscript fully available?

Reviewer #1: Yes

4. Is the manuscript presented in an intelligible fashion and written in standard English?

Reviewer #1: Yes

5. Review Comments to the Author

Reviewer #1: The topic of the article is interesting because finding substances that can influence the intestinal microbiota of piglets can be useful to avoid the development of intestinal diseases. Overall, the manuscript is well written, however, some points need to be reported more accurately. In particular, the paragraph on materials and methods should be written more clearly and completely. Especially, in the “Animal and housing” section, the creation of experimental groups should be described more clearly, defining the characteristics of the animals in trial, how the groups were balanced, the type of housing used, the way in which feed, and water are administered. Moreover, in the “Experimental design and sample collection” section, the modality of probiotic administration should be described in more detail, in particular regarding the time of administration. Also, unfortunately, some of the animals present in this study belong to another experiment which was described in a previous article that was actually under submission, and even if the data obtained in the previous research are not strictly connected to this work, some information about the animal, that could better specify some unclear point, is missing.

Title:

The title will be more appropriate by adding “infected piglets with ETEC E. coli”

Abstract:

When referring to statistically significant results the p-value is required.

Line 29: Please, be more precise “on five occasions” is not clear.

Line 29-30: Please add the supplemented dose

Line 30: The authors need to include the new nomenclature according to Zheng (10.1099/ijsem.0.004107)

Line 31: Based on what has been chosen the dose of 300 mg/Kg?

Line 33: It is not clear when they were challenged. How many days lasted the experimental trial?

Lines 34 – 36: Reword the sentence since as written it seems that the treatment done is exclusively nutritional

Line 36: Why did you decide to add chlortetracycline?

Line 43: Please, better specify what is meant by the term greater

Introduction:

Line 52: enterotoxigenic and verotoxigenic.

Line 54: antibiotics are more appropriate.

Line 80: Please use the new nomenclature for Lactobacillus genera.

Line 83: Which dose of Lactobacillus plantarum JDFM LP11?

Line 63: Please correct “microbiota of piglets, moreover”

Lines 71 – 73: The situation regarding antibiotic usage should refer to the global situation and not just Thailand.

Line 80: The authors need to include the new nomenclature according to Zheng (10.1099/ijsem.0.004107).

Line 82: Please substitute “use” with “absorption”.

Line 83: The authors need to include the new nomenclature according to Zheng (10.1099/ijsem.0.004107).

Lines 85 – 88: Better link the sentence on human studies being the study on piglets and having so far exposed only arguments in the veterinary field.

Line 86: Please add a reference.

Line 93: Better specify which features are considered safe and mention the reference.

Lines 93 – 94: How different strains of probiotics have demonstrated promising antibacterial, antiviral, anticonjugation, and antibiofilm action?

Line 94: If possible add other references that are not self-citations.

Line 95: Please, better define what the pronoun “them” refers to.

Line 96: How many CFU/g did you provide?

Material and methods:

Line 106: Please substitute “used” with “performed”

Line 112: Spell out the full name of the LAB acronym as it appears for the first time.

Line 115: Where are they balanced per weight and sex? Have you considered the litter effect?

Line 116 – 117: In the text, I would omit that the production and health data of the 60 pigs considered in the previous test were submitted elsewhere

Line 118: Better specify why you decide to add 12 piglets to the 60 already present in the previous experiment.

Lines 118 -119: How previous piglets were reared and handled? Please, better specify.

Lines 119 – 120: Please, reformulate the sentence, so written appears to be unclear.

Line 120: Please, better specify the experimental group’s composition. Are the six experimental groups balanced? How they are composed? Specify also the housing arrangements.

Line 123: Please move the sentences to lines 125 – 127 here.

Line 124: What were the procedures for transferring piglets from CPF Feed Research and Innovation Center to TFG Research Center? Can this affect the test results?

Line 129, Table S1: Provide the composition of “SP Premix”, substitute “salt” with sodium chloride, list the ingredients from most to less concentrated, provide the nutrient concentration for protein, fiber, and ashes.

Line 134: Please, better specify based on what the days of administration of probiotics were chosen.

Line 137: Describe the challenge procedure. Infection dosage, bacterial strain.

Line 138: Please, correct “were fed with a basal diet”

Line 141: Please, correct “were fed with a basal diet”

Line 143: Please, correct “were fed with a basal diet”

Lines 143 -151: Please, better specify when probiotic supplements were given, and how they were administered.

Line 154: Correct the typo. I understood that “hpc” stays for hours post-challenge and “dpc” for days post-challenge, however, the acronyms should be listed immediately close they in extenso form.

Statistical analysis is completely missing

Results:

It is not clear on which days the statistical differences have been observed.

All p-values are missing, for each statistically significant result a p-value has to be provided.

Write the following name in the same way both in the text and in the figure: tetW; tetQ; mefA; tetM.

Line 203, please delete “was” because it is repeated twice, “BacMet and was was accessed on 14-10-2019”.

Line 241-244: Provide the results of Shannon and Simpson indexes and their p-values.

Table 2: Please provide an error range and clearly indicate the statistically significant differences with lowercase letters and add the p-values.

Table 3: The same as mentioned before.

Line 249-S2 Figure: If you obtained significant differences for Bray-Curtis dissimilarity index please provide the p-value.

Line 268: Please, add the reference to the figure.

Line 291: Spell out the full name of AMR.

Line 356-357: Please list the investigated markers in the M&M section.

Table 4: Do you mean log2FC as normalized abundance?

Discussion:

Please discuss also the statistically significant results for diversity indexes.

Line 407-416: Data that are not presented or published elsewhere should not be discussed.

Line 460: Spell out the full name of WHO.

Line 472: Bacteroidaceae should be listed in italic.

Line 510: Avoid the use of “a lot”.

Line 532: Based on KEGG analysis you cannot state that they upregulated some genes but only speculate that they have higher potential for that particular metabolic pathway.

Line 535: Which amino acids?

Conclusions

Line 555: No results related to diarrhoea have been presented in this study

Line 558: No gene expression was performed in this study

6. PLOS authors have the option to publish the peer review history of their article (what does this mean?). If published, this will include your full peer review and any attached files.

Reviewer #1: No

---

## [Author Response · Author response to Decision Letter 0]

21 Apr 2022

Dear Editor of PLOS ONE

 Thank you very much for your suggestions to improve our manuscript " Metagenomic analysis of the gut microbiota in piglets either challenged or not with enterotoxigenic Escherichia coli reveals beneficial effects of probiotics on microbiome composition, resistome, digestive function and oxidative stress responses ". We appreciate your kind response and the reviewer’s extensive analysis. All suggestions have been carefully responded to topic by topic and noted in yellow outlining and blue lettering in the revised manuscript. We hope that our revisions will be satisfactory to merit publication in PLOS ONE. 

 Sincerely yours, 

 Prapasarakul N.

Corresponding author

Journal Requirements

1. Please ensure that your manuscript meets PLOS ONE's style requirements, including those for file naming

 We have checked throughout the manuscript and believed that this revised manuscript now meets the PLOS ONE style requirements.

2. In your Methods section, please include a comment about the state of the animals following this research. Were they euthanized or housed for use in further research? If any animals were sacrificed by the authors, please include the method of euthanasia, and describe any efforts that were undertaken to reduce animal suffering.

All the experimental piglets were sacrificed at the end of the experiment (at 42 days of age). We have provided the required information under “Animals and housing” (Page 5-6, Materials and methods section, Lines 117-121).

“This research was also supported by Pathogen Bank Project, Chulalongkorn Academic Advancement into Its 2nd Century Project and the CHE-TRF Senior Research Fund (RTA6280013).”

 Thank you for your comment. In the revised version, we have transferred the funding statement from the acknowledgment section into the funding section (Page 29, Funding section, Lines 588-594).

4. Please include captions for your Supporting Information files at the end of your manuscript, and update any in-text citations to match accordingly

 Thank you for your comment. We have revised about captions in the supporting information according to your recommendation.

Review Comments to the Author

1. Title: The title will be more appropriate by adding “infected piglets with ETEC E. coli”

 Thank you for your kind suggestion. In the revised manuscript, we have modified the title to “Metagenomic analysis of the gut microbiota in piglets either challenged or not with enterotoxigenic Escherichia coli reveals beneficial effects of probiotics on microbiome composition, resistome, digestive function and oxidative stress responses. This reflects the fact that three of the six experimental groups were challenged with ETEC.

2. Abstract: When referring to statistically significant results the p-value is required.

 Thank you for your comment. We totally agreed with your comment that where we state statistical significance in the results the p-value is necessary. However, statistical significance was not mentioned in the abstract. We explored the gut microbiome and resistome from pooled faecal samples of each experimental group and hence it was not appropriate to calculate statistical significance in this study.

3. Abstract: Line 29: Please, be more precise “on five occasions” is not clear.

 We are sorry that we did not make this issue clear. We have added the following additional information in the revised manuscript “On five occasions at 2, 5, 8, 11, and 14 days of piglet age” (Page 2, Abstract section, Line 30).

4. Abstract: Line 29-30: Please add the supplemented dose

Thank you for your comment. We have provided the required information in the revised version as “109 CFU/ml” (Page 2, Abstract section, Lines 31 and 33).

5. Abstract: Line 30: The authors need to include the new nomenclature according to Zheng (10.1099/ijsem.0.004107)

Thank you for your valuable suggestion. We have revised the bacterial nomenclature according to Zheng et al., 2020 into “Lactiplantibacillus” (Page 2, Abstract section, Line 32).

6. Abstract: Line 31: Based on what has been chosen the dose of 300 mg/Kg?

The administration of 300 mg/kg chlortetracycline in the feed is the therapeutic dosage which provides a growth promoting effect to weaning piglets according to the European Food Safety Authority (EFSA) recommendation [1, 2].

7. Abstract: Line 33: It is not clear when they were challenged. How many days lasted the experimental trial?

We are sorry for not making this clearer. The piglets were challenged at 28 days of age (7 days after weaning). Subsequently, they were held for another 14 days (to 42 days of age). These timings are now referenced at lines 29, 30 and 36.

8. Abstract: Lines 34 – 36: Reword the sentence since as written it seems that the treatment done is exclusively nutritional

We are sorry for not making this clear. We have revised the sentences according to your recommendation (Page 2, Abstract section, new Lines 37-40).

9. Abstract: Line 36: Why did you decide to add chlortetracycline?

Tetracycline is a broad-spectrum and low-cost antibiotic. Several tetracycline drugs such as chlortetracycline have various therapeutic indications related to several infections. Moreover, they are also frequently utilized in the production of food-producing animals, especially swine [3]. Hence, our study included administering chlortetracycline into the weaner diet of group 3 to mimic the commercial pig-rearing situation (added at Line 34).

10. Abstract: Line 43: Please, better specify what is meant by the term greater

Thank you for your valuable comment. We have revised the text in the revised manuscript to “with enrichment of genes” (Page 2, Abstract section, Line 45) and compared to other groups (line 46).

11. Introduction: Line 52: enterotoxigenic and verotoxigenic.

Thank you for your valuable comment. We have included verotoxigenic into the revised manuscript (Page 3, Introduction section, Line 54).

12. Introduction: Line 54: antibiotics are more appropriate.

Thank you for your kind suggestion. In the revised version, we have revised from medicine to antibiotics (Page 3, Introduction section, Line 56). 

13. Introduction: Line 63: Please correct “microbiota of piglets, moreover”

Thank you for your comment. We have corrected according to your kind suggestion (Page 3, Introduction section, Line 65).

14. Introduction: Lines 71 – 73: The situation regarding antibiotic usage should refer to the global situation and not just Thailand.

Thank you for your valuable comment. We have revised the text in the revised manuscript as “many countries, including Thailand” (Page 4, Introduction section, Line 74).

15. Introduction: Line 80: Please use the new nomenclature for Lactobacillus genera.

Thank you for your valuable suggestion. We have revised the bacterial nomenclature according to Zheng et al., 2020 into “Lactiplantibacillus” (Page 3, Introduction section, Line 83).

16. Introduction: Line 83: The authors need to include the new nomenclature according to Zheng (10.1099/ijsem.0.004107).

Thank you for your valuable suggestion. We have revised the bacterial nomenclature according to Zheng et al., 2020 into “Lactiplantibacillus” (Page 4, Introduction section, Line 86).

17. Introduction: Line 82: Please substitute “use” with “absorption”.

Thank you for your kind suggestion. In the revised version, we have revised from use to absorption (Page 4, Introduction section, Line 84).

18. Introduction: Line 83: Which dose of Lactobacillus plantarum JDFM LP11?

Thank you for your comment. We have provided additional information in the revised manuscript as “2.5×107 CFU/ml” (Page 4, Introduction section, Line 85).

19. Introduction: Line 91: The authors need to include the new nomenclature according to Zheng (10.1099/ijsem.0.004107).

Thank you for your valuable suggestion. We have revised the bacterial nomenclature according to Zheng et al., 2020 into “Lactiplantibacillus” (Page 4, Introduction section, Line 95). In addition, we have revised this new bacterial nomenclature throughout the revise version of manuscript.

20. Introduction: Lines 85 – 88: Better link the sentence on human studies being the study on piglets and having so far exposed only arguments in the veterinary field.

Thank you for your comment. We have provided more details about how the study of the pig resistome may be associating with human public health, according to your kind suggestion (Page 4, Introduction section, Line 92-94).

21. Introduction: Line 86: Please add a reference.

Thank you for your comment. We have now provided the required references in the revised manuscript with citation number 14 (Page 4, Introduction section, Line 89).

22. Introduction: Line 93: Better specify which features are considered safe and mention the reference.

Thank you for your comment. We have indicated more details about safety features and provided required references in the revised manuscript with citation number 16 (Page 5, Introduction section, Line 97-98).

23. Introduction: Lines 93 – 94: How different strains of probiotics have demonstrated promising antibacterial, antiviral, anticonjugation, and antibiofilm action?

Based on our previous studies, our probiotic strains comprising of Lactiplantibacillus plantarum strains 22F and 25F, and Pediococcus acidilactici strain 72N demonstrated all of the promising characteristics, including antibacterial, antiviral, anticonjugation, and antibiofilm activities as mentioned elsewhere [4-6].

24. Introduction: Line 94: If possible, add other references that are not self-citations.

Thank you for your comment. In this paragraph, we described the abilities of our probiotic strains. Thus, we needed to cite only our publications for supporting our hypothesis in this research that why these probiotic strains could modulate gut microbiome and resistome in piglets. These strains have not been used by others. 

25. Introduction: Line 95: Please, better define what the pronoun “them” refers to.

We are sorry that we did not make this clear. We have revised the pronoun “them” in the text into “our probiotic strains” (Page 5, Introduction section, Line 100).

26. Introduction: Line 96: How many CFU/g did you provide?

Thank you for your comment. We have provided more information as “used at a final concentration at 109 CFU/ml” (Page 5, Introduction section, Line 102).

27. Material and methods: Line 106: Please substitute “used” with “performed”

Thank you for your kind suggestion. In the revised version, we have revised from used to performed (Page 5, Material and methods section, Line 112).

28. Material and methods: Line 112: Spell out the full name of the LAB acronym as it appears for the first time.

Thank you for your kind suggestion. In the revised version, we have spelt out the full name of LAB (Page 6, Material and methods section, Line 122).

29. Material and methods: Line 115: Where are they balanced per weight and sex? Have you considered the litter effect?

 All experimental piglets were balanced for weight by using their birth weight (1.2-1.5 kg) and gender using stratified random sampling after the cross-fostering process.

We have considered the litter effect. Six healthy sows were included into our study. The piglets in each experimental group were equally assembled from those sows. Thus, the litter effects are unlikely to have influenced our results.

30. Material and methods: Line 116 – 117: In the text, I would omit that the production and health data of the 60 pigs considered in the previous test were submitted elsewhere

Thank you for your valuable comment, we have revised according to your kind suggestion, and we have provided the reference with citation number 22 (Page 6, Material and methods section, Line 126-127).

31. Material and methods: Line 118: Better specify why you decide to add 12 piglets to the 60 already present in the previous experiment.

In this research, the objective was set out to monitor and observed not only the gut microbiota but also the antibiotic resistome in the piglets. Hence, we decided to include an additional 12 piglets in the antibiotic group for use as positive control group of antibiotic administered piglets. We have provided these data (Page 6, Material and methods section, Line 127-128).

32. Material and methods: Lines 118 -119: How previous piglets were reared and handled? Please, better specify.

Thank you for your comment. We have provided the details about piglet rearing and handing in the “Animals and housing” section (Page 5-6, Material and methods section, Line 112-138).

33. Material and methods: Lines 119 – 120: Please, reformulate the sentence, so written appears to be unclear.

Thank you for your comment. We have revised the sentence according to your recommendation (Page 6, Material and methods section, Line 128).

34. Material and methods: Line 120: Please, better specify the experimental group’s composition. Are the six experimental groups balanced? How they are composed? Specify also the housing arrangements.

Information about the treatments received by the six experimental groups has been thoroughly described under “Experimental design and sample collection”. Briefly, piglets in groups 1-3 (the negative control, the probiotic control, and antibiotic groups) were not challenged with ETEC. At the same time, piglets in groups 4-6 (the single-strain, the multi-strain, and ETEC control groups) were challenged with ETEC stain L3.2 at 5x109 CFU/ml. (Page 7, Material and methods section, Line 145-163). In addition, the six experimental groups were balanced with male and female replicate pens per group (6 pigs per pen) (Page 6, Material and methods section, Line 129-130). For housing arrangements, we have provided the data in the “Animals and housing” (Page 6, Material and methods section, Line 132-138).

35. Material and methods: Line 123: Please move the sentences to lines 125 – 127 here.

Thank you for your comment. We have revised the text according to your kind suggestion (Page 6, Material and methods section, Line 132-135).

36. Material and methods: Line 124: What were the procedures for transferring piglets from CPF Feed Research and Innovation Center to TFG Research Center? Can this affect the test results?

All the experimental piglets were relocated by using animal transport vehicles that completely separated each of the experimental groups. After transferring to the TFG Research Center, all piglets were acclimatized for a week before performing ETEC challenge. Thus, the results in this study were not affected by stress arising from animal transfer.

37. Material and methods: Line 129, Table S1: Provide the composition of “SP Premix”, substitute “salt” with sodium chloride, list the ingredients from most to less concentrated, provide the nutrient concentration for protein, fiber, and ashes.

Thank you for your comment. We have provided the required information and revised the data according to your recommendation (Supporting information, Table S1).

38. Material and methods: Line 134: Please, better specify based on what the days of administration of probiotics were chosen.

Primary microbial colonizers are crucial for reconstituting the intestinal microbial community. Hence probiotic supplementation of neonatal piglets might be more effective for gut microbiota establishment, conferring several advantages to the health of pigs. The reasons for supplementing probiotic every 3 days were because we believed that probiotics need time for occupying the intestinal surfaces, and the supplementation on 5 occasions should enhance the numbers of successful colonized probiotics. All of these theories were stated in our previous study, and we have mentioned this in the revised manuscript (Page 6-7, Material and methods section, Line 142-144).

39. Material and methods: Line 137: Describe the challenge procedure. Infection dosage, bacterial strain.

Thank you for your comment. We have provided the required information as “All piglets in the three ETEC challenged group were orally inoculated with ETEC strain L3.2 at a final concentration at 5×109 CFU/ml at 28 days of age (7 days after weaning).” (Page 7, Material and methods section, Line 161-163).

40. Material and methods: Line 138: Please, correct “were fed with a basal diet”

Thank you for your kind suggestion. We have corrected as your recommendation (Page 6, Material and methods section, Line 148).

41. Material and methods: Line 141: Please, correct “were fed with a basal diet”

Thank you for your kind suggestion. We have corrected as your recommendation (Page 7, Material and methods section, Line 152-153).

42. Material and methods: Line 143: Please, correct “were fed with a basal diet”

Thank you for your kind suggestion. We have corrected as your recommendation (Page 7, Material and methods section, Line 154).

43. Material and methods: Lines 143 -151: Please, better specify when probiotic supplements were given, and how they were administered

Thank you for your comment. We have indicated the time for probiotic supplementation as “the three groups supplemented with probiotics received these on five occasions, when the piglets were 2, 5, 8, 11, and 14 days of age” (Page 6-7, Material and methods section, Line 142-144). Moreover, we have provided the required information about the probiotic supplementation procedure (Page 7, Material and methods section, Line 149-160).

44. Material and methods: Line 154: Correct the typo. I understood that “hpc” stays for hours post-challenge and “dpc” for days post-challenge, however, the acronyms should be listed immediately close they in extenso form.

Thank you for your kind suggestion. We have corrected as per your recommendation (Page 7-8, Material and methods section, Line 166-167).

45. Material and methods: Statistical analysis is completely missing

46. Results: It is not clear on which days the statistical differences have been observed. 

47. Results: All p-values are missing, for each statistically significant result a p-value has to be provided.

Thank you for your valuable comment. As explained in the discussion section, statistical analysis was not conducted in this research because “samples were pooled because it was not technically or financially possible to examine samples from all individual piglets in this study. It is acknowledged that this does not allow comparison of variations between pigs within a group, but this approach was necessary for practical purposes and does provide an overview of group affects. The methodology used means that it was not appropriate to undertake statistical analysis between groups in this study.” (Page 22-23, Discussion section, Line 443-448). Therefore, statistical differences or p-values were not calculated throughout the current study. 

48. Results: Write the following name in the same way both in the text and in the figure: tetW; tetQ; mefA; tetM.

Thank you for your comment. We have revised all figures according to your kind suggestion throughout the revised manuscript.

49. Results: Line 203, please delete “was” because it is repeated twice, “BacMet and was was accessed on 14-10-2019”.

Thank you for your comment. We have revised according to your kind suggestion (Page 10, Material and methods section, Line 214).

50. Results: Line 241-244: Provide the results of Shannon and Simpson indexes and their p-values.

51. Results: Table 2: Please provide an error range and clearly indicate the statistically significant differences with lowercase letters and add the p-values.

52. Results: Table 3: The same as mentioned before.

53. Results: Line 249-S2 Figure: If you obtained significant differences for Bray-Curtis dissimilarity index please provide the p-value.

Thank you for your valuable comment. As explained above, statistical analysis was not conducted in this research due to the nature of the sampling.

54. Results: Line 268: Please, add the reference to the figure.

Thank you for your comment. We have added the reference to the figure according to your kind suggestion (Page 13, Results section, Line 286).

55. Results: Line 291: Spell out the full name of AMR.

Thank you for your kind suggestion. In the revised version, we have spelt out the full name of AMR (Page 14, Results section, Line 309).

56. Results: Line 356-357: Please list the investigated markers in the M&M section.

Thank you for your kind suggestion. In the revised version, we have provided the required data according to your recommendation (Page 11, Materials and methods section, Line 242-245).

57. Results: Table 4: Do you mean log2FC as normalized abundance?

Data within Table 4 are normalized abundance which was generated by MG-RAST using DESeq analysis, and we have provided this information in the revised manuscript (Page 11, Materials and methods section, Line 245-247). 

58. Discussion: Please discuss also the statistically significant results for diversity indexes.

Thank you for your valuable comment. As detailed above we did not attempt statistical comparisons because of the nature of the (pooled) samples.

59. Discussion: Line 407-416: Data that are not presented or published elsewhere should not be discussed.

Thank you for your comment. Data that we employed in the discussion section had been recently accepted for publication in Scientific Reports. Moreover, we have now provided the required reference in the revised manuscript with citation number 22 following the format of accepted article in the submission guideline of PLOSONE (Page 22, Discussion section, Line 425).

60. Discussion: Line 460: Spell out the full name of WHO.

Thank you for your kind suggestion. In the revised version, we have spelt out the full name of WHO (Page 24, Discussion section, Line 479).

61. Discussion: Line 472: Bacteroidaceae should be listed in italic.

Thank you for your comment. We have revised according to your kind suggestion (Page 24, Discussion section, Line 491).

62. Discussion: Line 510: Avoid the use of “a lot”.

Thank you for your comment. We have revised according to your kind suggestion (Page 26, Discussion section, Line 529-530).

63. Discussion: Line 532: Based on KEGG analysis you cannot state that they upregulated some genes but only speculate that they have higher potential for that particular metabolic pathway.

Thank you for your comment. We have revised according to your kind suggestion (Page 27, Discussion section, Line 551).

64. Discussion: Line 535: Which amino acids?

Thank you for your kind suggestion. We have provided additional data in the revised manuscript (Page 27, Discussion section, Line 554-555).

65. Conclusions: Line 555: No results related to diarrhoea have been presented in this study

Thank you for your comment. We have revised according to your kind suggestion (Page 28, Discussion section, Line 573).

66. Conclusions: Line 558: No gene expression was performed in this study

Thank you for your comment. We have revised according to your kind suggestion (Page 28, Conclusion section, Line 576).

References

1. Hazards EPoB, Koutsoumanis K, Allende A, Alvarez-Ordonez A, Bolton D, Bover-Cid S, et al. Maximum levels of cross-contamination for 24 antimicrobial active substances in non-target feed. Part 12: Tetracyclines: tetracycline, chlortetracycline, oxytetracycline, and doxycycline. EFSA J. 2021;19(10):e06864. Epub 2021/11/04. doi: 10.2903/j.efsa.2021.6864. PubMed PMID: 34729092; PubMed Central PMCID: PMCPMC8546800.

2. McEvoy JD, Crooks SR, Elliott CT, McCaughey WJ, Kennedy DG. Origin of chlortetracycline in pig tissue. Analyst. 1994;119(12):2603-6. Epub 1994/12/01. doi: 10.1039/an9941902603. PubMed PMID: 7879861.

3. Ghanbari M, Klose V, Crispie F, Cotter PD. The dynamics of the antibiotic resistome in the feces of freshly weaned pigs following therapeutic administration of oxytetracycline. Sci Rep. 2019;9(1):4062. doi: 10.1038/s41598-019-40496-8. PubMed PMID: 30858509; PubMed Central PMCID: PMCPMC6411716.

4. Apiwatsiri P, Pupa P, Yindee J, Niyomtham W, Sirichokchatchawan W, Lugsomya K, et al. Anticonjugation and antibiofilm evaluation of probiotic strains Lactobacillus plantarum 22F, 25F, and Pediococcus acidilactici 72N against Escherichia coli harboring mcr-1 gene. Front Vet Sci. 2021;8:614439. Epub 2021/06/29. doi: 10.3389/fvets.2021.614439. PubMed PMID: 34179153; PubMed Central PMCID: PMCPMC8225926.

5. Sirichokchatchawan W, Pupa P, Praechansri P, Am-In N, Tanasupawat S, Sonthayanon P, et al. Autochthonous lactic acid bacteria isolated from pig faeces in Thailand show probiotic properties and antibacterial activity against enteric pathogenic bacteria. Microb Pathog. 2018;119:208-15. doi: 10.1016/j.micpath.2018.04.031. PubMed PMID: 29678738.

6. Sirichokchatchawan W, Temeeyasen G, Nilubol D, Prapasarakul N. Protective effects of cell-free supernatant and live lactic acid bacteria isolated from Thai pigs against a pandemic strain of porcine epidemic diarrhea virus. Probiotics Antimicrob Proteins. 2018;10(2):383-90. doi: 10.1007/s12602-017-9281-y. PubMed PMID: 28434154.

---

## [Editor Report · Decision Letter 1]

1 Jun 2022

Metagenomic analysis of the gut microbiota in piglets either challenged or not with enterotoxigenic Escherichia coli reveals beneficial effects of probiotics on microbiome composition, resistome, digestive function and oxidative stress responses

PONE-D-22-04406R1

Dear Dr. Prapasarakul,

We’re pleased to inform you that your manuscript has been judged scientifically suitable for publication and will be formally accepted for publication once it meets all outstanding technical requirements.

Kind regards,

Horacio Bach

Academic Editor

PLOS ONE
---

## [Editor Report · Acceptance letter]

8 Jun 2022

PONE-D-22-04406R1 

Metagenomic analysis of the gut microbiota in piglets either challenged or not with enterotoxigenic Escherichia coli reveals beneficial effects of probiotics on microbiome composition, resistome, digestive function and oxidative stress responses 

Dear Dr. Prapasarakul:

I'm pleased to inform you that your manuscript has been deemed suitable for publication in PLOS ONE. Congratulations! Your manuscript is now with our production department. 

Kind regards, 

on behalf of

Prof. Horacio Bach 

Academic Editor

PLOS ONE